# Acute Rubigine^®^ Poisoning in Martinique a French Overseas Department of America: Clinical Characteristics and Prognostic Factors

**DOI:** 10.3390/toxics10080453

**Published:** 2022-08-05

**Authors:** Jonathan Florentin, Remi Neviere, Papa Gueye, Bruno Mégarbane, Hossein Mehdaoui, Dabor Resiere

**Affiliations:** 1Service des Urgences Adultes, Centre Hospitalier Universitaire de Martinique (Fort-de-France), CEDEX, 97261 Fort-de-France, France; 2Service de Réanimation Polyvalente, Centre Hospitalier Universitaire de Martinique (Fort-de-France), CEDEX, 97261 Fort-de-France, France; 3SAMU Centre 15, Centre Hospitalier Universitaire de Martinique (Fort-de-France), CEDEX, 97261 Fort-de-France, France; 4Soins Intensifs Médicaux et Toxicologiques, Hôpital Lariboisière, Université Paris-Diderot, 75011 Paris, France

**Keywords:** Rubigine^®^, hydrofluoric acid, ammonium bifluoride, acute poisoning, Martinique

## Abstract

Rubigine^®^ is an anti-rust stain remover containing fluorides which is believed to have been the cause of many deaths in Martinique. However, after the modification of its composition in 2006, serious poisoning from old formulas containing fluorides persisted. Our main objective was to determine the clinical characteristics and prognostic factors of these intoxications. Methods: Any patient admitted to the Martinique University Hospital for acute Rubigine^®^ poisoning was included from 1 January 2000 to 31 December 2016. Usual demographic and clinical data were collected and comparisons between surviving and deceased patients made using a univariate analysis and logistic regression. Results: Fifty-five patients were included (mean age: 43 years; sex ratio M/F: 1.1), and the main clinical characteristics were: changes in electrocardiogram (ECG) (80%), digestive system disorders (75%), and neurological disorders (12%). The main features linked to death were the presence of hydrofluoric acid (*p* < 0.0001), age over 55 years (*p* = 0.01), hypocalcemia after the initial intravenous calcium supplementation (*p* = 0.0003), diarrhea (*p* < 0.0001), hypersialorrhea (*p* < 0.0001), myocardial excitability (*p* < 0.0001), and state of shock (*p* < 0.0001). Three patients required circulatory support by venous-arterial ECMO. Mortality was 10.9%. Conclusions: Rubigine^®^ poisoning is responsible for significant morbidity and mortality. Fortunately, its incidence as well as mortality has sharply decreased in Martinique thanks to the measures taken by the French state. This retrospective work nevertheless shows that acute intoxication by the old formula of Rubigine^®^ remains the main factor of poor prognosis.

## 1. Introduction

Hydrofluoric acid (HF) poisoning is a medical emergency and represents a significant public health problem since it is fatal in nearly 50% of cases [1,2,3]. Rubigine^®^ is an acid caustic product (pH < 1) composed of fluorides; it is a very common rust remover used in the Caribbean, more particularly in Martinique. It has been reported that, in the 1960s, it was the cause of a large number of deaths [4]. Its composition was not modified until April 1994, when a prefectural decree was published relating to the declaration, classification, packaging, and labeling of substances [5]. This decree allowed for a spectacular reduction in mortality related to the ingestion of this product. Before this decree, Rubigine^®^ was presented in the form of a white 100-mL bottle with a green cap containing 10% HF, 10% ammonium bifluoride (AB) (i.e., about 20 g [1 mole] of HF), and 1% oxalic acid. From 1994 to 2006, the Rubigine^®^ product contained 10% AB (i.e., about 11.5 g of HF) and 1% oxalic acid. Today, its composition is no longer toxic, and its ingestion leads to an irritation of the mucosa; it contains water, ammonia hydroxide, and oxalic acid (pH < 3).

However, in a study published in 2020 by Resiere et al. [6], it was noted that Rubigine^®^ poisoning alone represented 8 to 10% of cases of severe poisoning in Martinique in the period from 2000 to 2010.

HF and AB are strong caustic products; their ingestion is responsible for liquefactive necrosis lesions and sometimes even gastrointestinal perforation. However, they are more specifically known for their very severe cardiac and metabolic toxicity, due in part to their ability to chelate calcium and magnesium, forming insoluble fluorapatite ([3(Ca_3_(PO_4_)2Ca(F_2_)] salts that are responsible for hypocalcemia and hypomagnesemia, as well as their ability to inhibit various metalloenzymes including different metabolic chains (carbohydrate metabolism, cellular respiratory chain, Krebs cycle, anaerobic glycolysis, etc.) [7,8].

Standard management of poisoning due to skin contact with HF is well established [9]. However, the mechanisms and management of poisoning due to ingestion of HF can still be improved [8]. Therefore, our main objective was to determine the clinical characteristics and prognostic factors of these cases of poisoning. Our secondary objective was to establish a protocol that will be used for the multidisciplinary management of Rubigine^®^ poisoning.

## 2. Methods

We have carried out a retrospective study based on files collected over a 16-year period from January 2000 to December 2016, including all patients admitted to the emergency room and intensive care unit of CHU de Martinique (Martinique University Hospital Center) for acute Rubigine^®^ poisoning. Patients admitted to the emergency room and intensive care unit of CHU de Martinique, with a primary or secondary diagnosis of acute Rubigine^®^ poisoning, were selected from the files of the Medicinal Information Department (Département d’Information Médicale—DIM) using the Centaure software of SAMU 972 (the dedicated Medical Rescue Team of the CHU de Martinique). The data collected from the files were grouped into several areas of focus: demographic and epidemiological characteristics, characteristics of the poisoning, clinical and electrocardiographic characteristics, biological characteristics, and care data.

## 3. Statistical Analysis

The data are presented in the form of median (25th–75th percentiles) for quantitative data or percentages for qualitative data. A univariate analysis was carried out using Mann–Whitney tests for quantitative data and Fisher’s tests for qualitative data. A multivariate analysis by logistic regression was carried out. A *p*-value of less than 0.05 was considered significant.

The statistical analysis was performed using JMP Statistical software (SAS).

## 4. Results

A total of 55 patients were included. There were 29 male patients and 26 female patients, including one 15-year-old child. The mean age was 43 years (+/−14.2), the minimum age was 15 years, and the maximum age was 82 years. In 54 cases, the intoxication was intentional. The death rate was 10.9% and the survival rate was 89.1%; the prevalence of acute poisoning and death is shown in Figure 1.

All patients presented clinical symptoms including a change in electrocardiogram (ECG) in 79% of cases, epigastric pain (75%), pharyngeal pain (58%), vomiting (53%), wide and pointed T waves (40%), ST segment depression (36%), hematemesis (25%), wide QRS (21%), hypersialorrhea (20%), diarrhea (20%), acute renal failure (20%), state of shock (16%), fasciculations (13%), coma (13%), ventricular rhythm disorders (11%), myoclonus (7%), long QT (4%), paresthesia (3%), seizures (3%), and gastrointestinal perforation (2%).

Hypocalcemia after the first intravenous calcium supplementation was the most common biological involvement (41%), while hypercalcemia after supplementation (5%) and hyperkalemia (2%) were rarely observed. In addition, patients also had a decreased alkaline reserve (31%), acidosis (29%), elevated lactates (21%), and/or coagulation problems (15%). The description of the population is presented in Table 1, and the results of the univariate analysis of the relationship between clinical-biological characteristics and the occurrence of death are presented in Table 2.

In our study, pre-hospital management consisted primarily of stabilizing the patient, immediately starting supplementation via intravenous injection of calcium (64%), which allowed for ECG improvement in 15 patients (28%). Digestive decontamination was performed for 35 patients (63%). At the hospital, the resuscitation care most commonly performed was orotracheal intubation (22%), external electric shocks (11%), renal replacement therapy (11%), hemodynamic support by adrenaline (7%), noradrenaline (4%), and dobutamine (37%), as well as peripheral circulatory support by venoarterial ECMO (6%).

We also performed a logistic regression comparing survival and death, whose model explains 60% of the variance for the following characteristics: age, heart rate, and occurrence of diarrhea. The area under the ROC curve was 0.97 (Table 3).

## 5. Discussion

For several decades, the prognostic characteristics and management of poisoning due to HF ingestion has been a major public health problem due to its irregular occurrence and the lack of a consensus conference [1,7,10]. However, this poisoning is becoming increasingly common due to the great versatility of HF and its use in an increasing number of industries [9,11]. Absorption of HF at the stomach is very rapid, and its main toxic action occurs at a biological level, with severe chelating and metabolic effects [8,12].

Our study shows that age over 55 years, diarrhea, and hypersialorrhea are significantly associated with occurrence of death, as we can observe in the literature [2,13]. Likewise, the presence of myocardial excitability must be considered a factor of poor prognosis: the existence of negative T waves, long QT, and QRS are the most common changes; however, cardiac involvement leads to death by ventricular fibrillation and refractory cardiogenic shock [1,2,13].

In addition, vomiting and pain in the upper gastrointestinal tract are the digestive symptoms most commonly found [1,10]. A single patient from our study had a gastrointestinal perforation, which was not associated with death.

In our series, we noted a wide QRS in 12 patients, which was associated with hypotension in four patients. Is this a membrane-stabilizing effect? In any case, it seems logical to mention it in the face of any acute fluoride poisoning with cardiac involvement. However, no clinical study has clearly made the connection between this poisoning and a membrane-stabilizing effect [14].

Different studies have shown that hypocalcemia is the most common biological consequence in this type of poisoning [15,16]. Hyperkalemia is also a disorder frequently found in severe fluoride poisoning [2,3,8,17]. In our study, a single patient presented with hyperkalemia at 6 mmol/L. Acidosis is more of a prognostic marker. In our study, it was metabolic and lactic acidosis. On average, lactate levels are 15 mmol/L in deceased patients, which, for us, clearly illustrates the severe metabolic impairment in acute Rubigine^®^ poisoning.

In HF poisoning, the lethal dose is estimated at 20 mg/kg [18]; this corresponds to an estimated quantity of product between 16 and 20 mL (in adults) for a solution composed of 10% HF. Our study shows that the supposed ingested dose frequently exceeds 10 times the lethal dose, namely 160 mL of a solution containing 10% AB, and 100 mL of a solution containing 10% HF and 10% AB; however, 89% of patients survived, unlike the series described in the literature, in which nearly 50% of patients died [1,10]. In our study, deaths are exclusively observed in cases of poisoning with formulas containing 10% HF. The epidemiological analysis shows that the last death corresponds to the last poisoning by the old formula of Rubigine^®^; in total, we observed six deaths in the 16 years analyzed, including four in the year 1993 [19]. This shows that the change in law was more effective than therapeutic management in significantly reducing the morbidity and mortality due to this poisoning. However, it has been proven that early treatment of hypocalcemia and hypomagnesemia is correlated with a better prognostic factor for survival [15,20,21,22].

The medical regulation phase consists of providing first aid advice to callers and engaging suitable means of transport as quickly as possible. The supposed dose ingested and the time of the onset of toxidrome are essential in the initial questioning; once the diagnosis is confirmed, the patient’s main characteristics must be determined, and the patient urgently transferred to a suitable critical care service.

During pre-hospital management, the airways must be stabilized and secured, fluid therapy with crystalloid solutions must be initiated, and the patient’s level of consciousness must be assessed. After that, it is necessary to look for cardiac and digestive clinical prognostic signs. During transport, an initial ECG must be performed, followed by continuous cardiac monitoring of the patient. We recommend starting supplementation with an injection of 2 mg of magnesium sulfate and 4 g of calcium gluconate in 500 mL of saline via slow administration (about 1 h); calcium chloride could cause overly rapid changes in blood calcium, which could lead to other types of heart problems. Digestive decontamination is not beneficial in the management of this type of poisoning. Treatment for electrocardiographic disorders involves correcting as best possible any electrolyte imbalances, acidosis, and other metabolic disorders. Hospitalization in critical care is essential for the management of these poisoned patients. In the presence of an electrical storm, other techniques or exceptional treatments must be used, because repeated external electric shocks do not make it possible to reduce ventricular fibrillation; various studies have not shown any benefit in the use of this technique [1,13,23].

HF is a caustic agent that can cause gastrointestinal necrotic lesions as well as perforations [8]. This is why some authors recommend performing an upper gastrointestinal endoscopy early, specifically in the first 12 h following poisoning [24]. Unfortunately, in Martinique, few patients have benefited from this due to the lack of a well-established protocol and consensus. However, we have observed only one single case of digestive perforation, which was not fatal. In addition, some authors have proposed renal replacement therapy as a therapeutic alternative, particularly with the goal of rapidly correcting electrolyte imbalances and clearing the fluoride ion from the body [25,26]. We have not used this therapy early in acute Rubigine^®^ poisoning. In our analysis, 50% of the patients who underwent renal replacement therapy died. French toxicology reference works and some studies that have indicated that overly rapid and abrupt correction of electrolyte disorders may be harmful and could promote the onset of other ventricular rhythm disorders [8,12,14]. Renal replacement therapy in the early management of acute Rubigine^®^ poisoning did not significantly reduce the number of deaths in our study. This technique should be used according to known and validated recommendations in intensive care medicine, particularly in cases of hyperkalemia [27].

Peripheral circulatory support was offered for the first time in our series. The three patients who underwent VA-ECMO due to refractory cardiogenic shock and cardiac arrest refractory to conventional treatments all died following organ failures or massive hemorrhages. In addition to being a highly corrosive acid, fluoride avidly binds many cations, and therefore scavenges trace elements critical to the normal functioning of many enzyme systems. By binding calcium, fluoride can disrupt and arrest many enzymatic and intracellular processes, such as signaling pathways, ionic transport, energy metabolism, and redox status [28]. Mitochondria are the key intracellular targets for fluoride. Fluoride ions can bind to functional amino acid groups surrounding the active center of an enzyme to cause an inhibitory effect, as is the case for the enzymes of the glycolytic pathway and the Krebs cycle [29]. In addition, Na+/K+-ATPases are also inhibited, leading to ATP depletion and a cell energetic crisis. Fluoride also acts as an uncoupling agent that induces the opening of the permeability transition pore and loss of mitochondrial membrane potential. In addition, fluoride can induce the release of cytochrome c (Cyt C) from the mitochondria to the cytosol, activating apoptotic pathways [28,29]. Eventually, fluoride has been shown to induce oxidative stress resulting from the excessive production of ROS at the mitochondrial level. In our experience, circulatory assistance has not demonstrated efficacy in the management of severe Rubigine^®^ poisoning.

## 6. Conclusions

Rubigine^®^ poisoning is responsible for significant morbi-mortality. Fortunately, its incidence has greatly decreased in Martinique. However, this work shows that acute poisoning by the old Rubigine^®^ formula is the main factor of poor prognosis, regardless of the level of intensive care treatment. In addition, older age and the presence of early gastrointestinal and cardiac signs both seem to be clinical characteristics of poor prognosis; this is why we recommend initiating treatment with intravenous calcium and magnesium as soon as possible in the case of any acute fluoride poisoning in order to prevent the majority of deaths. However, it is the change in laws concerning these products that has reduced mortality.

## Figures and Tables

**Figure 1 toxics-10-00453-f001:**
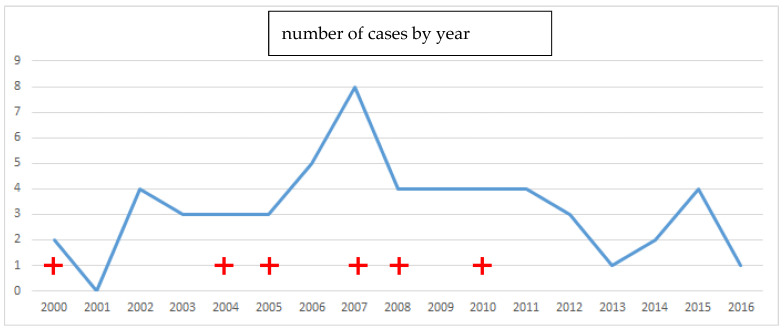
Prevalence curve of deaths (red cross) and cases of acute Rubigine^®^ poisoning in Martinique.

**Table 1 toxics-10-00453-t001:** Description of the population analyzed.

Characteristics	Survivors *n* = 49 Median [Percentiles]	Deaths *n* = 6 Median [Percentiles]
Age (years)	42 [34–48]	62 [44–70]
Duration of hospitalization (days)	2 [2–4]	1 [1–1.25]
Duration of intubation (days)	0 [0–0]	1 [1–1]
Heart rate (bpm)	80 [70–90.5]	107 [89–120]
Minimum systolic BP (mmHg)	118 [95.7–152]	60 [50–121]
Blood potassium (mmol/L)	3.8 [3.5–4]	5.3 [4.3–5.6]
Creatinine (µmol/L)	90 [74–105]	103.5 [98–157.2]
Corrected calcium after IV supplementation (mmol/L)	2.2 [2–2.4]	1.6 [1.4–1.8]
Blood phosphorus (mmol/L)	1.1 [0.9–1.3]	1.3 [1.3–1.4]
Blood magnesium (mmol/L)	0.8 [0.7–0.9]	0.3 [0.2–0.9]
Alkaline reserve (mmol/L)	23 [20.7–25]	16.5 [15–17.2]
Total proteins (G/L)	72.5 [66.7–77.2]	58 [48.5–65]
White blood cells (Giga/L)	6.3 [5.2–7.6]	10.5 [9.5–11.6]
Platelets (giga/L)	229 [196–273]	165 [137.5–170.5]
Lactates (mmol/L)	2 [1.6–2.8]	16.5 [10.6–20.2]
pH	7.39 [7.34–7.40]	7.30 [6.8–7.38]
PaO_2_ (mmHg)	99 [90–136]	258 [159.5–311]
PT (%)	80 [74–90.5]	53 [47–55]
Fibrinogen (G/L)	3.4 [2.8–3.5]	5.6 [5.5–6.5]
AST (IU/L)	22 [18–35.7]	136 [108–167.5]
ALT (IU/L)	20.5 [15.5–30]	65 [49.2–85]

BP: blood pressure; PaO_2_: partial pressure of oxygen; PT: prothrombin time; AST: aspartate aminotransferase; ALT: alanine aminotransferase.

**Table 2 toxics-10-00453-t002:** Univariate analysis of relationship between clinical-biological characteristics and occurrence of death.

Characteristics	Survival n (%)	Mean	Death n (%)	Mean	*p*
Age (years)		41 ± 13		60 ± 15	0.01
Duration of hospitalization (days)		4 ± 5		1 ± 0	0.001
Duration of intubation (days)		1 ± 3		1 ± 0	<0.0001
VA ECMO	0		3 (50)		<0.0001
Adrenaline	0		4 (67)		<0.0001
Dobutamine	0		2 (33)		<0.0001
Intubation	6 (12.6)		6 (100)		<0.0001
Cardiac history	4 (6)		2 (30)		0.02
State of shock	3 (6)		6 (100)		<0.0001
Myocardial excitability	2 (4)		6 (100)		<0.0001
Heart rate (bpm)		83 ± 17		104 ± 16	0.007
Minimum systolic BP (mmHg)		117 ± 23		80 ± 43	0.04
Hypersialorrhea	5 (11)		5 (83)		<0.0001
Diarrhea	5 (11)		5 (83)		<0.0001
Gastrointestinal perforation	1 (2)		0		0.7
Coma	5 (10)		2 (33)		0.1
Acute renal failure	7 (14)		4 (66.6)		0.0028
Renal replacement therapy	3 (6)		3 (50)		0.0013
Creatinine (µmol/L)		97 ± 41		124 ± 40	0.02
Presence of HF	0		6 (100)		<0.0001
Hypocalcemia despite supplementation	16 (32)		6 (100)		0.0017
Corrected calcium after IV supplementation (mmol/L)		2.2 ± 0.4		1.6 ± 0.2	0.0003
Blood magnesium (mmol/L)		0.83 ± 0.16		0.53 ± 0.40	0.03
Blood potassium (mmol/L)		3.7 ± 0.3		5.0 ± 0.7	0.0009
Blood phosphorus (mmol/L)		1.1 ± 0.3		1.4 ± 0.1	0.04
pH		7.38 ± 0.04		7.17 ± 0.28	0.04
Lactates (mmol/L)		2.9 ± 2.8		15.2 ± 6.0	0.0003
Alkaline reserve (mmol/L)		23.0 ± 2.9		16.0 ± 1.2	0.0002
PaO_2_ (mmHg)		120 ± 48		239 ± 97	0.03
PT (%)		81 ± 14		51 ± 4	0.001
Fibrinogen (G/L)		3.3 ± 0.8		5.9 ± 0.6	0.0008
AST (IU/L)		35 ± 36		135 ± 57	0.0005
ALT (IU/L)		28.4 ± 35.9		66.8 ± 21.6	0.0006
Total proteins (G/L)		72 ± 9		57 ± 9.53	0.004
White blood cells (Giga/L)		6.8 ± 2.7		10.5 ± 1.1	0.0034
Platelets (giga/L)		236 ± 59		156 ± 21.9	0.0017

BP: blood pressure; PaO_2_: partial pressure of oxygen; PT: prothrombin time; AST: aspartate aminotransferase; ALT: alanine aminotransferase.

**Table 3 toxics-10-00453-t003:** Logistic regression based on age, heart rate, and diarrhea.

Characteristics	*p*	Area under the ROC Curve
Age	0.01	0.97
Heart rate	0.007	
Diarrhea	<0.0001	

## Data Availability

Not applicable.

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
