# Peer review of "Acute Rubigine® Poisoning in Martinique a French Overseas Department of America: Clinical Characteristics and Prognostic Factors"

_toxics, 2022, doi:10.3390/toxics10080453_

Round 1

Reviewer 1 Report

This is a very interesting study looking, retrospectively, at the toxicity of hydrofluoric acid from poisoning by old formulas of Rubigine®.

The population of patients is very well described along with the signs of toxicity and treatment (hypocalcemia). The impact of the changes in regulation and formulation of this product on these dreadful intoxications is also well described. The discussion is well documented. I have only few comments that the authors may want to consider.

1-    It seems that the lethality is the consequence of a severe shock with hyperlactacidemia (La >10 mM). The "cardiogenic" nature of the shock is suspected and probably real, but there is no additional information, in the table or the text, confirming that the shock was cardiogenic. Can the authors provide any additional data supporting the nature of the shock (vaoplegic vs cardiogenic) based on hemodynamic measurements or echocardiography (this information must be available in the patients who were treated with ECMO)? If these data are not available, could the authors comment and discuss if other mechanisms of shock (vasoplegic) can be produced by this intoxication.

2-    There is a significant elevation of blood La in the lethal forms of intoxication. The authors mention in their introduction possible metabolic consequences of  intoxications by hydrofluoric acid , i.e. inhibition of the glycolytic activity, the Krebs cycle or even mitochondrial electron chain via interaction with metalloenzymes. This point is not presented in the discussion section. What additional information or data from the literature could be provided by the authors to support such a mechanism of toxicity? Do the author believe that the (cardiogenic) shock is the consequence of this inhibition of metabolism or a direct effect of a severe hypocalcemia? Is the severity of the hyerlactacidemia a consequence of the shock or of a potential metabolic dysfunction (depression of the Krebs cycle or electron chain)? It seems that the victims do not to present major neurological symptoms (coma, seizure, apnea) commonly seen with metabolic intoxication. Is that correct? Could the author comment and briefly, discuss these few points which could be of interest for many readers.

Author Response

Reviewer #1

Comments and Suggestions for Authors

This is a very interesting study looking, retrospectively, at the toxicity of hydrofluoric acid from poisoning by old formulas of Rubigine®.

The population of patients is very well described along with the signs of toxicity and treatment (hypocalcemia). The impact of the changes in regulation and formulation of this product on these dreadful intoxications is also well described. The discussion is well documented. I have only few comments that the authors may want to consider.

  • It seems that the lethality is the consequence of a severe shock with hyperlactacidemia (La >10 mM). The "cardiogenic" nature of the shock is suspected and probably real, but there is no additional information, in the table or the text, confirming that the shock was cardiogenic. Can the authors provide any additional data supporting the nature of the shock (vaoplegic vs cardiogenic) based on hemodynamic measurements or echocardiography (this information must be available in the patients who were treated with ECMO)? If these data are not available, could the authors comment and discuss if other mechanisms of shock (vasoplegic) can be produced by this intoxication.

Response to the reviewer:

Response 1. We thank the reviewer for the helpful comments on the issue of the type of shock and hemodynamic measurements to characterize the shock.

The ingestion of solutions during involuntary and intentional exposures to hydrofluoric acid not only causes skin burns, eye damage, and digestive and respiratory disorders but also causes systemic toxicity of fluorides, including cardiovascular symptoms; in the most severe cases, electrolyte imbalance and enzymatic inhibition occur, leading to cardiac arrhythmias such as ventricular fibrillation, and also, cardiac arrest, cardiogenic shock, myocarditis, and death.Fluoride ions bind calcium and magnesium, which can occur at a rate that exceeds the body's ability to mobilize calcium and magnesium in the serum. In the majority of cases, hemodynamic evaluation is performed by electrocardiography for QT prolongation and arrhythmias (secondary to hypocalcemia), and also by transthoracic echocardiography (TTE, usually global hypokinesia,  left ventricular ejection fraction less than 10%), and the patient should be placed on cardiac monitoring. The pathophysiological mechanism is multifactorial; it is well thought that fluoride ions can cause hypocalcemia, and it is also thought that fluoride ions are directly toxic to myocardial cells by inhibiting adenylate.

This type of shock is mixed; on the one hand, we have a vasoplegic shock resulting from hydro electrolytic disorders causing important metabolic disorders (hypocalcemia and hypomagnesemia); on the other hand, the direct attack of fluorides on the myocardial cells causes a rhythmic storm (ventricular fibrillation), cardiac arrest leading to refractory cardiogenic shock requiring circulatory assistance ECMO VA to ensure the tissue perfusion.

Due to the persistence of a rhythmic storm and a refractory cardiogenic shock state, the patient was placed on circulatory assistance by Extracorporeal membrane oxygenation (ECMO) veno-arterial (VA). Despite the ECMO in place, the patient presented several episodes of Ventricular fibrillation and rapidly evolved towards a state of multi-visceral failure, leading to the patient's death, despite these heavy and exceptional therapeutic means.

  • There is a significant elevation of blood La in the lethal forms of intoxication. The authors mention in their introduction possible metabolic consequences of  intoxications by hydrofluoric acid , i.e. inhibition of the glycolytic activity, the Krebs cycle or even mitochondrial electron chain via interaction with metalloenzymes. This point is not presented in the discussion section. What additional information or data from the literature could be provided by the authors to support such a mechanism of toxicity? Do the author believe that the (cardiogenic) shock is the consequence of this inhibition of metabolism or a direct effect of a severe hypocalcemia? Is the severity of the hyerlactacidemia a consequence of the shock or of a potential metabolic dysfunction (depression of the Krebs cycle or electron chain)? It seems that the victims do not to present major neurological symptoms (coma, seizure, apnea) commonly seen with metabolic intoxication. Is that correct? Could the author comment and briefly, discuss these few points which could be of interest for many readers.

Response 2. We agree with the reviewer about the lack of information on mitochondria effects. We added 2 references.

In addition to being a highly corrosive acid, fluoride avidly binds many cations and therefore scavenges trace elements critical to the normal functioning of many enzyme systems. By binding calcium, fluoride can disrupt and arrest many enzymatic and intracellular processes, such as signaling pathways, ionic transport, energy metabolism, and redox status [29]. Mitochondria are the key intracellular targets for fluoride. Fluoride ions can bind to functional amino acid groups surrounding the active center of an enzyme to cause an inhibitory effect, as is the case for enzymes of the glycolytic pathway and the Krebs cycle [30]. In addition, Na+ /K+ -ATPases are also inhibited, leading to ATP depletion and cell energetic crisis. Fluoride also acts as an uncoupling agent that induces the opening of the permeability transition pore and loss of mitochondrial membrane potential. In addition, fluoride can induce the release of cytochrome c (Cyt C) from the mitochondria to the cytosol, activating apoptotic pathways [29, 30]. Eventually, fluoride has been shown to induce oxidative stress resulting from excessive production of ROS at the mitochondrial level. In our experience, circulatory assistance has not demonstrated efficacy in the management of severe Rubigine® poisoning.

  1. Barbier O, Arreola-Mendoza L, Del Razo LM. Molecular mechanisms of fluoride toxicity. Chem Biol Interact. 2010,188(2), 319-33.
  2. Araujo TT, Barbosa Silva Pereira HA, Dionizio A, Sanchez CDC, de Souza Carvalho T, da Silva Fernandes M, Rabelo Buzalaf MA. Changes in energy metabolism induced by fluoride: Insights from inside the mitochondria. Chemosphere. 2019, 236,124357.

Reviewer 2 Report

This is an interesting report concernic Rubigine poisoning in Martinique during 2000-2016. I am interested if the poisonings were accidental or intentional. Did you analyse this, if it is possible it is worthy of inclusion in the manuscript.

In the author list I find Papa Gueye whose contribution is not listed in the end of the manuscript, but his name is included in the acknowledgments. So please decide if this is an author (and add his contribution) or just leave his name in Acknowledgements and delete his name from the list of authors.

Author Response

Reviewer #2

Comments and Suggestions for Authors

This is an interesting report concernic Rubigine poisoning in Martinique during 2000-2016. I am interested if the poisonings were accidental or intentional. Did you analyse this, if it is possible it is worthy of inclusion in the manuscript.

Response 2. We agree with the reviewer regarding the circumstances of the ingestion. A total of 55 patients were included. There were 29 male patients and 26 female patients, including one 15-year-old child. The mean age was 43 years (+/- 14.2), the minimum age was 15, and the maximum age was 82. In 54 cases, the intoxication was intentional.

In the author list I find Papa Gueye whose contribution is not listed in the end of the manuscript, but his name is included in the acknowledgments. So please decide if this is an author (and add his contribution) or just leave his name in Acknowledgements and delete his name from the list of authors.

Response 2. We agree with the reviewer regarding Papa Gueye’s participation. Pr Gueye participated actively in the investigation. Therefore, we decide to leave his name as a co-author. We added then Yannick Brouste instead of Papa Gueye.
